# Associations of Metabolic and Obstetric Risk Parameters with Timing of Lactogenesis II

**DOI:** 10.3390/nu14040876

**Published:** 2022-02-19

**Authors:** Amber J. Mullen, Deborah L. O’Connor, Anthony J. Hanley, Giovanni Piedimonte, Maeve Wallace, Sylvia H. Ley

**Affiliations:** 1Department of Epidemiology, Tulane University School of Public Health and Tropical Medicine, 1440 Canal Street, M.B. 8318, New Orleans, LA 70112, USA; amullen1@tulane.edu; 2Department of Nutritional Sciences, University of Toronto, Toronto, ON M5S 1A8, Canada; deborah.oconnor@utoronto.ca (D.L.O.); anthony.hanley@utoronto.ca (A.J.H.); 3Translational Medicine Program, The Hospital for Sick Children, Toronto, ON M5G 0A4, Canada; 4Dalla Lana School of Public Health, University of Toronto, Toronto, ON M5T 3M7, Canada; 5Division of Endocrinology, University of Toronto, Toronto, ON M5S 3H2, Canada; 6Leadership Sinai Centre for Diabetes, Mount Sinai Hospital, Toronto, ON M5T 3L9, Canada; 7Departments of Pediatrics, Biochemistry and Molecular Biology, Tulane University School of Medicine, New Orleans, LA 70112, USA; gpiedimonte@tulane.edu; 8Department of Social, Behavioral, and Population Sciences, Tulane University School of Public Health and Tropical Medicine, New Orleans, LA 70112, USA; mwallace@tulane.edu

**Keywords:** delayed onset of lactogenesis II, DLII, pregnancy, lactation, breastfeeding, human milk, metabolic, obstetric, mother, infant

## Abstract

Lactogenesis II is the onset of copious milk production following parturition. Delayed onset of lactogenesis II (DLII) often contributes to poorer lactation performance, which may adversely affect maternal and child health. The present study aims to identify the metabolic and obstetric risk factors for DLII in a secondary analysis of a prospective cohort study following pregnant women through postpartum. We defined the onset of lactogenesis II as delayed if it occurred ≥72 h postpartum. Multiple logistic regression analyses were conducted to evaluate the associations of metabolic and obstetric variables with DLII. Median onset of lactogenesis II was 72.4 h (IQR 60.4–91.6) postpartum, and 55.4% (98 of 177) of women experienced DLII. Time to first breast contact ≥ 2 h postpartum compared to ≤1 h postpartum was associated with DLII (OR 2.71 95% CI 1.12–6.53) with adjustment for age, race, pregravid BMI, primiparity, and mode of delivery, while metabolic variables were not significantly associated with DLII. In this comprehensive examination of potential metabolic and obstetric parameters, earlier timing of putting the infant to the breast remained significantly associated with earlier onset of milk coming in after consideration of the other potential risk factors. Obstetrical practices, including putting the baby to the breast later, may have an important impact on the timing of lactation, and interventions are needed to address this concern.

## 1. Introduction

The World Health Organization (WHO), Canadian Paediatric Society (CPS), and American Academy of Pediatrics (AAP) recommend initiation of breastfeeding within one hour of birth, exclusive breastfeeding for the first six months of life, and continued breastfeeding for two years or beyond [1,2,3]. Breast milk is the optimal food source for newborns as it contains not only nutrients but also an array of bioactive substances that are essential for infant growth and long-term development [4,5]. Moreover, lactation is associated with lower later life risk of many cardiometabolic outcomes and cancers among women [6,7,8,9].

Lactogenesis is the process by which the mammary glands develop the ability to secrete milk [10]. It takes place in two stages: secretory differentiation (lactogenesis I) and secretory activation (lactogenesis II) [11]. Lactogenesis II occurs following parturition and is characterized by the onset of copious milk production [12,13]. This stage generally begins between 48 and 72 h postpartum, but its timing can be influenced by numerous endocrine, metabolic, and obstetric factors [14,15,16,17,18]. Delayed onset of lactogenesis II (DLII) is defined as the onset of lactogenesis ≥72 h postpartum [6,15,17,18,19]. The incidence of DLII has ranged between 22% and 44% in various populations [20]. DLII is associated with early reduction and cessation of breastfeeding, which undermines the documented benefits of lactation for the mother and the infant [14,17,18,21]. 

One of the most well-established risk factors for DLII is primiparity [15,16,17,21]. “Milk coming in” often corresponds to degree of parity, with multiparas sensing it sooner than primiparas [12]. Evidence also suggests that maternal body mass index (BMI) and gestational diabetes mellitus (GDM) are associated with DLII [15,16,17,19,22]. Mothers with metabolic abnormalities, such as obesity and diabetes, have been shown to experience poorer lactation performance due to mechanical difficulties and reduced milk hormone concentrations [23,24,25]. Other cited risk factors include maternal age and mode of delivery, with women of older age and women who deliver by cesarean section at increased risk for DLII, respectively [14,16,21]. Time to first breastfeeding has been implicated as a risk factor for DLII as well, but the significance of its impact remains considerably understudied [15,21].

It is critical to identify the risk factors for DLII so that women experiencing these risk factors receive appropriate care during pregnancy and parturition and are provided with additional support postpartum. The objective of the present study is to examine the potentially modifiable metabolic and obstetric risk variables associated with DLII.

## 2. Materials and Methods

### 2.1. Study Design and Population

The present study is a secondary data analysis of a prospective cohort study involving women and their offspring. Pregnant women were recruited from outpatient clinics between April 2009 and July 2010 at Mount Sinai Hospital, a large tertiary care center in Toronto, Ontario, Canada where more than 7,000 pregnancies are followed annually [4]. Signed informed consent was obtained from 271 pregnant women, and the study was approved by the Mount Sinai Hospital Research Ethics Board. This study is registered at clinicaltrials.gov as NCT01405547.

The findings from this cohort have been previously published [26,27,28,29,30,31]. For the current secondary analysis, we analyzed data from late pregnancy to one week postpartum. The inclusion criterion was women aged 20 years or older who intended to breastfeed. The exclusion criterion was women who reported pre-existing type 1 diabetes or type 2 diabetes (T2DM). Those with incomplete 3 h oral glucose tolerance test (OGTT) data, i.e., missing metabolic variable data (*n* = 56), and those missing lactogenesis data (*n* = 38) were excluded. Therefore, a total of 177 pregnant women were included (Figure 1).

### 2.2. Data Collection 

Interviewers administered baseline questionnaires during pregnancy to obtain demographic and medical history information. Height and weight were measured following standardized anthropometric protocols [32]. At the time of the study, universal screening for GDM in all pregnant women between 24 and 48 weeks gestation was standard obstetrical practice in Canada. A glucose challenge test (GCT) was ordered to measure plasma glucose concentration 1 h after ingestion of a 50 g glucose load [33]. If the plasma glucose concentration was ≥7.8 mmol/L, the patient was referred for a diagnostic OGTT, in which plasma glucose values were measured while fasting and then measured hourly for 3 h following ingestion of 100 g of glucose. In clinical practice, the OGTT would generally only be ordered if the GCT was abnormal, whereas in this study the baseline pregnancy OGTT was completed in all participating women.

The OGTT provided four categories of maternal glucose tolerance in pregnancy [34]: women with GDM (defined as having two or more of the following [35]: fasting glucose ≥5.8 mmol/L, 1 h blood glucose ≥10.6 mmol/L, 2 h blood glucose ≥9.2 mmol/L, and 3 h blood glucose ≥ 8.1 mmol/L); women with an OGTT indicating gestational impaired glucose tolerance (defined as exceeding only one of the aforementioned glycemic thresholds [34]); women with an abnormal GCT but a normal OGTT (defined as exceeding none of the aforementioned glycemic thresholds); and women with a normal GCT and a normal OGTT.

Medical information related to delivery and birth was obtained for the mother and the infant from a hospital clinical database, as well as interviewer-administered questionnaires including lactogenesis II related questions during the first week postpartum.

### 2.3. Biochemical Procedures and Analyses

Maternal serum samples were processed and analyzed for glucose and insulin concentrations according to established protocols at the Banting and Best Diabetes Centre laboratory, Mount Sinai Hospital, Toronto, Ontario, Canada. Aliquots of serum and plasma were prepared and frozen immediately at −80 °C for additional assays. Insulin sensitivity (ISogtt) and beta-cell function (HOMA-IR) were assessed by validated indices derived from insulin and glucose measurements during the OGTT [36,37].

### 2.4. Timing of Lactogenesis II

Information on DLII was obtained during the first week postpartum. Within the first 3 days postpartum, women were asked to recall the nearest hour when they started to feel their milk come in. Specifically, mothers were asked to recall the presence of breast “tingling”, fullness, and swelling. This technique for assessing the onset of lactogenesis II by maternal perception has been validated with test weighting [38]. If the mother had not experienced lactogenesis II by day 3 postpartum, the woman was contacted via telephone on day 7 postpartum (±2 days). If the mother still reported no milk coming in by day 7, the time of the onset of lactogenesis II was recorded as greater than 7 days postpartum. DLII was defined as the onset of lactogenesis II ≥72 h postpartum [6,15,17,18,19].

### 2.5. Exposure Variables

A variety of metabolic and obstetric variables suspected to influence DLII were investigated. These variables included: age, race and ethnicity, pregravid BMI, family history of T2DM, GDM, serum concentrations at OGTT, gestational age, primiparity, mode of delivery, infant birth weight, and time to first breast contact. Age was defined as date of birth. Race and ethnicity was recorded as either White, Black, Asian, South Asian, Indigenous Canadian, or Other, and was subsequently categorized as White or non-White. Pregravid BMI was calculated [weight (kg)/height (m^2^)] using the self-reported pregravid weight and standardized height measures collected at baseline. Family history of T2DM was defined as the self-reported confirmation of such on the baseline questionnaire. GDM was defined as having two or more of the following [35]: fasting glucose ≥5.8 mmol/L, 1 h blood glucose ≥10.6 mmol/L, 2 h blood glucose ≥9.2 mmol/L, and 3 h blood glucose ≥8.1 mmol/L). Serum concentrations at OGTT included fasting glucose, insulin resistance (HOMA-IR), insulin sensitivity (ISogtt), and adiponectin [26]. Gestational age was defined as the number of weeks from conception until birth. Primiparity was defined as giving birth for the first time. Mode of delivery was categorized as scheduled cesarean, unscheduled cesarean, or spontaneous. Infant birth weight was examined by a medical professional postpartum and measured to the nearest 0.01 kg, and then categorized into tertiles. Time to first breast contact was defined as the number of hours until the first infant breastfeeding attempt.

### 2.6. Statistical Analyses

We assessed distributions of the continuous variables for normality, and transformations of the skewed variables were used in the statistical analyses as appropriate. Descriptive statistics for the continuous variables were summarized as mean ± standard deviation or as median (25th–75th percentile) for the variables with a skewed distribution. The categorical variables were summarized using proportions. Characteristics of the study population were stratified by DLII status. Bivariate differences between the groups were compared using t-test for continuous data and chi-square test for categorical data. We performed multiple logistic regression analyses to assess the association between maternal metabolic and obstetric measures and DLII, with adjustment for potential confounders. The outcome variable was the presence of DLII, and the main exposure variables were maternal metabolic and obstetric measures. The following models were constructed based on the findings of previous studies, including the variables of biological and obstetrical relevance. Model 1 was adjusted for age and race. Model 2 was additionally adjusted for primiparity and pregravid BMI. We also performed analysis for a model which simultaneously adjusted for the aforementioned variables (age, race, primiparity, and pregravid BMI), as well as for the potential confounders identified in the bivariate analysis (*p* ≤ 0.10) (mode of delivery and time to first breast contact). Interactions of primiparity or pregravid BMI with the exposure variables were also tested. Odds ratios (ORs) were used to indicate the risk for DLII. All *p*-values were two-sided, and significance was set at alpha < 0.05. The Statistical Analysis System (SAS), Version 9.4 (SAS Institute, Cary, NC, USA) was used to analyze all data.

## 3. Results

### 3.1. Study Population Characteristics

Between March 2009 and July 2010, 271 pregnant women were recruited. Of those recruited, 94 were ineligible for the current investigation because of incomplete 3 h OGTT data (*n* = 56) or missing lactogenesis data (*n* = 38). Participating mothers (*n* = 177) were predominantly White (58.2%) primiparas (52.5%) who delivered spontaneously (60.2%). The average maternal age was 34.8 years (4.3 SD). The median gestational age was 39.1 weeks (IQR 38.3–40.3). Overall, the average onset of lactogenesis II was 78.9 h postpartum (31.0 SD) and the median onset of lactogenesis II was 72.4 h postpartum (IQR 60.4–91.6).

### 3.2. Association of Metabolic and Obstetric Variables with DLII

The characteristics of the current study population are presented by DLII status (Table 1). Over half of the participating mothers (55.4%) experienced DLII. Among them, the average timing of the onset of lactogenesis II was 98.3 h postpartum (30.1 SD). Bivariate analysis also identified the following obstetric measures as potential confounders eligible for inclusion in multivariable analysis (*p* ≤ 0.10): primiparity, mode of delivery, and time to first breast contact. That is, compared to mothers who did not experience DLII, the mothers who did experience DLII were more likely to be primiparous, have undergone an unscheduled cesarean, and have first breastfed ≥ 2 h postpartum. There were no statistically significant bivariate associations between any of the metabolic measures and DLII.

The effects of mode of delivery and time to first breast contact were statistically significant in bivariate analysis and continued to be statistically significant after further multivariable analyses (Table 2). Unscheduled cesarean and time to first breast contact ≥ 2 h postpartum were identified as significant risk factors for DLII when adjusted for age and race. After additional adjustment for primiparity and pregravid BMI, the association of unscheduled cesarean with DLII became attenuated and no longer significant. The association between time to first breast contact and DLII increased after additional adjustment for primiparity and pregravid BMI. No statistically significant associations were found between any metabolic measures and DLII (Table 3). No statistically significant interactions between any of the exposure variables and primiparity or pregravid BMI were detected.

The third multiple logistic regression model included age, race, pregravid BMI, primiparity, mode of delivery, and time to first breast contact (Table 4). In this model, time to first breast contact was the sole statistically significant independent predictor of DLII, with ≥2 h postpartum until first breast contact conferring nearly three times greater risk for DLII.

## 4. Discussion

The present study examined the impact of a variety of metabolic and obstetric measures on the timing of the onset of lactogenesis II. Among 177 women recruited during pregnancy, those who subsequently experienced DLII were more likely to be primiparous, have undergone an unscheduled cesarean, and have first breastfed ≥ 2 h postpartum. Unscheduled cesarean and time to first breast contact ≥ 2 h postpartum were associated with higher risk of DLII after adjustment for age and race. Furthermore, time to first breast contact ≥ 2 h postpartum independently predicted DLII after adjustment for age, race, pregravid BMI, primiparity, and mode of delivery.

Consistent with the known risk factors, we found that DLII occurred more often among primiparous women than multiparous women [15,16,17,21]. Additionally, in alignment with previous research, we found that women who underwent an unscheduled cesarean were more likely to experience DLII than women who delivered spontaneously (OR 2.24 95% CI 1.02–4.92) [14,18,21]. These results support the evidence indicating that DLII is an increasingly common problem among primiparas and that obstetrical practices, such as unscheduled cesarean, may place pregnant women at greater risk for DLII. Primiparity and unscheduled cesareans are not modifiable, due to the respective biological and emergent nature, but women with these risk factors should be prioritized for enhanced postpartum care and breastfeeding support.

Our results also identify one lesser researched risk factor as exerting the most significant effect. Women whose first breastfeeding occurred ≥ 2 h postpartum were more likely to experience DLII than women whose first breastfeeding occurred ≤ 1 h postpartum (OR 2.43 95% CI 1.09–5.45). Moreover, time to first breast contact was the sole statistically significant independent predictor of DLII after adjustment for all other variables in the final multiple logistic regression model (OR 2.71 95% CI 1.12–6.53). Time to first breast contact is a relatively understudied risk factor for DLII, so future directions should include more research on this exposure and on subsequent infant growth [15,21]. Furthermore, current obstetrical practices rightly focus on proper positioning and attachment of the infant to the breast, but these practices should also focus on ensuring that this occurs as soon as possible, or at least prior to 2 h postpartum, to reduce the burden of DLII and facilitate optimal lactation [39,40,41].

While lactogenesis II generally occurs between 48 and 72 h postpartum, the women in this study reported an average onset of 78.9 h postpartum (31.0 SD). The incidence of DLII was 55.4%, which is considerably higher than that reported for women in similar populations [20]. This finding may be because the original study was designed to investigate the risk factors for T2DM. As such, many individuals at higher risk for T2DM participated in the study, although at risk for T2DM was not an inclusion criterion. The onset of lactogenesis II among those that experienced DLII occurred at an average of 98.3 h postpartum (30.1 SD). We did not detect any significant bivariate or multivariate associations between the metabolic measures and DLII. Taken together, our findings suggest that obstetric factors may have a greater impact on the timing of the onset of lactogenesis II than metabolic factors. To prevent cases of DLII, public health and obstetric interventions that support the establishment and maintenance of lactation should be implemented. Such interventions should especially target the modifiable risk factor of time to first breast contact by promoting breastfeeding shortly after birth and frequently thereafter.

As with all observational studies, our study had limitations. First, we relied on maternal perception to obtain information on the timing of the onset of lactogenesis II. While this technique has been validated, it remains a subjective assessment. Second, the entire study population was recruited from Mount Sinai Hospital. Doing so may have minimized variability in maternity care practices and, in turn, some of our obstetric variables. Third, use and type of anesthesia were not included among our exposure variables. The original study upon which this secondary analysis is based preceded the literature on the potential effect of general anesthesia versus epidural anesthesia on the onset of lactogenesis II [42,43,44]. These data were therefore not collected and unable to be considered in the present study. Fourth, examining the influence of supplementation was not an objective of the present study. Future studies that further examine the metabolic and obstetric factors among those who breastfeed exclusively compared to those who supplement with formula may enable a more distinctive understanding of the role of maternal physiology and the role of breastfeeding management in DLII.

## 5. Conclusions

We found that the timing of the onset of lactogenesis II was associated with primiparity, mode of delivery, and time to first breast contact. Women who were primiparous, underwent an unscheduled cesarean, or did not breastfeed until ≥2 h postpartum were significantly more likely to experience DLII. These findings reinforce many known risk factors for DLII. However, few studies have examined time to first breast contact as a risk factor for DLII. Our study underscored the significance of the early postpartum period for the success of breastfeeding, specifically that lactation success was strongly influenced by when breastfeeding was initiated. Obstetrical practices, including putting the baby to the breast later, may have an important impact on the timing of lactation, and interventions are needed to address this concern.

## Figures and Tables

**Figure 1 nutrients-14-00876-f001:**
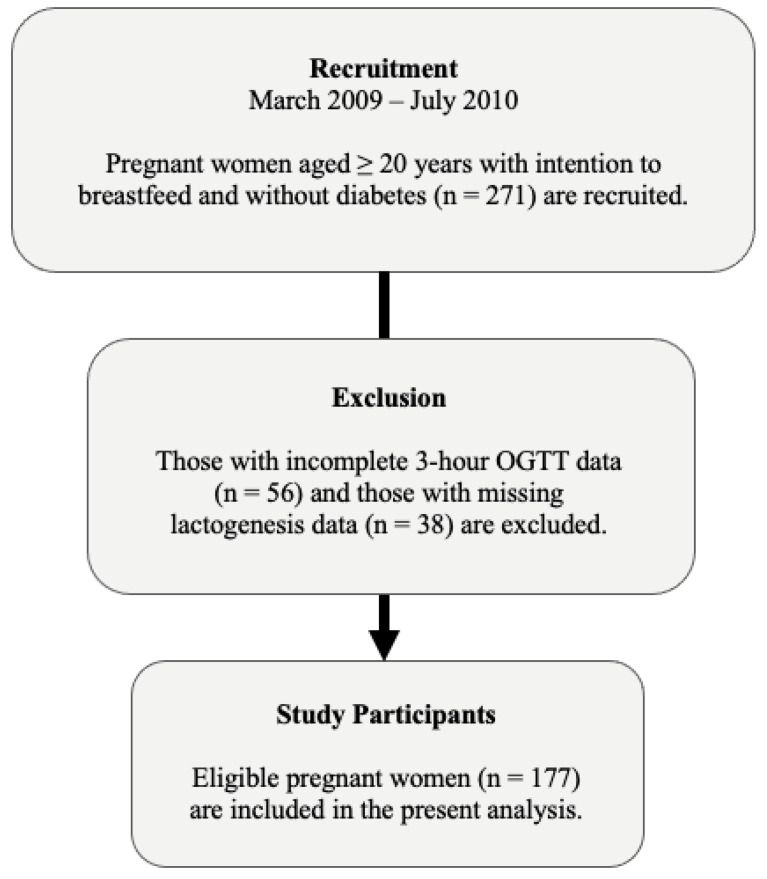
Flow diagram of study participants.

**Table 1 nutrients-14-00876-t001:** Characteristics of the study population by delayed onset of lactogenesis II status.

Risk Factor	DLII—Yes *n* (%) or Mean ± SD	DLII—No *n* (%) or Mean ± SD
Age (y)	34.7 ± 4.5	35.0 ± 4.2
Race		
White	53 (54.1)	50 (63.3)
Non-white	45 (45.9)	29 (36.7)
Pregravid BMI (kg/m^2^)	24.8 ± 4.6	25.0 ± 5.5
Family history of T2DM		
Yes	53 (54.1)	42 (53.2)
No	45 (45.9)	37 (46.8)
Gestational diabetes		
Yes	24 (24.5)	16 (20.3)
No	74 (75.5)	63 (79.7)
Serum concentrations at OGTT		
Fasting glucose (mmol/L)	4.7 ± 0.6	4.7 ± 0.9
HOMA-IR	2.3 ± 1.4	2.3 ± 1.6
ISogtt ^1^	10.9 (8.3, 15.8)	13.0 (8.3, 18.2)
Adiponectin (μg/mL)	7.1 ± 2.4	7.4 ± 2.2
Gestational age (wk)	39.3 ± 1.2	39.0 ± 1.2
Primiparous		
Yes	60 (61.2)	33 (41.8)
No	38 (38.8)	46 (58.2)
Mode of delivery		
Scheduled cesarean	10 (10.2)	17 (21.8)
Unscheduled cesarean	31 (31.6)	12 (15.4)
Spontaneous	57 (58.2)	49 (62.8)
Infant birth weight (kg)	3.4 ± 0.5	3.4 ± 0.5
Time to first breast contact (h)	3.8 ± 7.0	2.3 ± 4.5
Time to onset of lactogenesis II (h)	98.3 ± 30.1	57.2 ± 11.7

^1^ Median (25th, 75th percentiles). T2DM, type 2 diabetes; OGTT, oral glucose tolerance test; HOMA-IR, homeostatic model assessment for insulin resistance; ISogtt, insulin sensitivity index for oral glucose tolerance tests.

**Table 2 nutrients-14-00876-t002:** Logistic regression model (obstetric measures) estimating odds of delayed onset of lactogenesis II.

Obstetric Variable	OR (95% CI) ^1^	OR (95% CI) ^2^
Mode of delivery	0.52 (0.21–1.26)	0.62 (0.24–1.57)
Scheduled cesarean	2.24 (1.02–4.92)	1.93 (0.86–4.35)
Unscheduled cesarean Spontaneous	1.00 [Reference]	1.00 [Reference]
Infant birth weight		
<3.2 kg	0.98 (0.46–2.07)	1.03 (0.47–2.23)
3.2 kg–3.6 kg	1.00 [Reference]	1.00 [Reference]
>3.6 kg	1.21 (0.58–2.54)	1.20 (0.56–2.56)
Time to first breast contact		
≤1 h	1.00 [Reference]	1.00 [Reference]
1.1–2 h	1.35 (0.62–2.96)	1.30 (0.57–2.97)
≥2 h	2.30 (1.09–4.86)	2.43 (1.09–5.45)

^1^ Model 1. Adjusted for age and race. ^2^ Model 2. Adjusted for age, race, primiparity, and pregravid BMI.

**Table 3 nutrients-14-00876-t003:** Logistic regression model (metabolic measures) estimating odds of delayed onset of lactogenesis II.

Metabolic Variable	OR (95% CI) ^1^	OR (95% CI) ^2^
Family history of T2DM		
Yes	1.02 (0.55–1.87)	1.03 (0.55–1.92)
No	1.00 [Reference]	1.00 [Reference]
Gestational diabetes		
Yes	1.21 (0.57–2.57)	1.26 (0.58–2.73)
No	1.00 [Reference]	1.00 [Reference]
Fasting glucose	0.92 (0.60–1.41)	0.97 (0.62–1.52)
HOMA-IR	0.99 (0.80–1.22)	1.05 (0.82–1.35)
ISogtt	0.99 (0.96–1.02)	0.99 (0.95–1.02)
Adiponectin	0.97 (0.84–1.11)	0.96 (0.83–1.11)

^1^ Model 1. Adjusted for age and race. ^2^ Model 2. Adjusted for age, race, primiparity, and pregravid BMI. T2DM, type 2 diabetes; HOMA-IR, homeostatic model assessment for insulin resistance; ISogtt, insulin sensitivity index for oral glucose tolerance tests.

**Table 4 nutrients-14-00876-t004:** Logistic regression model (simultaneously adjusting for metabolic and obstetric measures) estimating odds of delayed onset of lactogenesis II.

Variable	OR (95% CI) ^1^
Age (y)	0.99 (0.91–1.08)
Race	
White	1.00 [Reference]
Non-White	1.29 (0.66–2.53)
Pregravid BMI	0.98 (0.92–1.05)
Primiparity	
Yes	1.96 (0.92–4.18)
No	1.00 [Reference]
Mode of delivery	
Scheduled cesarean	0.48 (0.17–1.33)
Unscheduled cesarean	1.42 (0.60–3.40)
Spontaneous	1.00 [Reference]
Time to first breast contact	
≤1 h	1.00 [Reference]
1.1–2 h	1.34 (0.57–3.17)
≥2 h	2.71 (1.12–6.53)

^1^ Model 1. Adjusted for all other variables in the model.

## Data Availability

Data described in the manuscript, code book, and analytic code will be made available upon reasonable request.

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
