# Peer review of "Associations of Metabolic and Obstetric Risk Parameters with Timing of Lactogenesis II"

_nutrients, 2022, doi:10.3390/nu14040876_

Round 1

Reviewer 1 Report

Introduction

- This work addresses a topic of interest for the promotion and support of breastfeeding, the association between early initiation of breastfeeding and lactogenesis II. However, this is not new. There is previous evidence of this association and its relationship with better rates of breastfeeding. For this reason, the recommendation of early initiation of breastfeeding (immediately after delivery) has been included in clinical practice guidelines on breastfeeding (*) and in the Ten Steps to Successful Breastfeeding of the Baby-Friendly Hospital Initiative, from UNICEF (**). These references should be included in the introduction or in the discussion section of the manuscript.

(*) Perinatal Services BC Health Promotion Guideline. Breastfeeding Healthy Term Infants. May 2012 (v3 March 2015). Available in: http://www.perinatalservicesbc.ca/search?k=breastfeedinghealthyterminfantguideline.pdf

Guía de práctica clínica sobre lactancia maternal. Ministerio de Sanidad, Servicios Sociales e Igualdad. 2017. Available in: https://portal.guiasalud.es/gpc/lactancia-materna/

(**) Baby-Friendly Hospital Initiative. Ten steps to successful breastfeeding, from UNICEF and the World Health Organization. Available in: https://www.unicef.org/documents/baby-friendly-hospital-initiative

- Page 2: The text of lines 58 and 59 could be deleted since it repeats what was already expressed in lines 55-57.

Materials and Methods

- Page 2, line 78: Explain why more than 10 years have passed since the collection of the data (April 2009 – July 2010) until its analysis

- Page 2, lines 86-88, 94-96 and figure 1. Did all the women participating in the study have a glucose challenge test (GCT) with a glucose concentration greater than or equal to 7.8 mmol/L after one hour of ingesting 50 g of glucose?. It is indicated that this was the criterion for referred the patient for the 3-hour oral glucose tolerance test (lines 94-96) and that 56 mothers with incomplete 3-hour OGTT data (lines 86-87) were excluded from the study. This information should be clarified.

- Page 2, lines 90-92: If the weight and height measurements were taken during pregnancy, how were you able to calculate the pregravid BMI? Was it really the pregravid BMI or the BMI in the first months of pregnancy?

-Page 3. Exposure variables (lines 130-148).

  • Why were the use of anesthesia and the type of anesthesia during delivery not included among the exposure variables?. Previous studies show that it could delay the onset of lactogenesis II (&) The inclusion of this variable would allow comparing the effect of general anesthesia versus epidural anesthesia on the onset of lactogenesis II.

(&) Lind JN, Perrine CG, Li R. Relationship between Use of Labor Pain Medications and Delayed Onset of Lactation. J Hum Lact. 2014 

Kutlucan L, Seker Ä°S, Demiraran Y, Ersoy Ö, Karagöz Ä°, Sezen G, Köse SA. Effects of different anesthesia protocols on lactation in the postpartum period. J Turk Ger Gynecol Assoc. 2014 Dec 1;15(4):233-8.

Kocaöz FÅž, Destegül D, Kocaöz S. Comparison of the breastfeeding outcomes and self-efficacy in the early postpartum period of women who had given birth by cesarean under general or spinal anesthesia. J Matern Fetal Neonatal Med. 2019 Jul 10:1-5

  • It is not indicated how the variable "time to first breast contact" was registered. If it came from the hospital clinical database or from the interviewer-administered questionnaires

Results

- Table 1 should include p-values from bivariate analyses.

- In the logistic regression analyzes (Tables 2, 3 and 4), the need for various models is not adequately justified. The last logistic regression model (table 4) simultaneously adjusting for metabolic and obstetric measures, including those of model 1 (only adjusted for age and race) and model 2 (adjusted for age and race, already included in model 1, and primiparity and pregravid BMI )

Discussion

Page 8,  lines 253-254.  The phrase “This finding is likely because the current cohort population is at higher risk for T2DM” should be explained, and information on this characteristic of the sample should be included in the “Study Design and Population” of the Material and Methods section.

Conclusions

The text of this section is very extensive. It should be summarized in a paragraph.

Author Response

Reviewer 1:

1)  This work addresses a topic of interest for the promotion and support of breastfeeding, the association between early initiation of breastfeeding and lactogenesis II. However, this is not new. There is previous evidence of this association and its relationship with better rates of breastfeeding. For this reason, the recommendation of early initiation of breastfeeding (immediately after delivery) has been included in clinical practice guidelines on breastfeeding (*) and in the Ten Steps to Successful Breastfeeding of the Baby-Friendly Hospital Initiative, from UNICEF (**). These references should be included in the introduction or in the discussion section of the manuscript.

(*) Perinatal Services BC Health Promotion Guideline. Breastfeeding Healthy Term Infants. May 2012 (v3 March 2015). Available in: http://www.perinatalservicesbc.ca/search?k=breastfeedinghealthyterminfantguideline.pdf

Guía de práctica clínica sobre lactancia maternal. Ministerio de Sanidad, Servicios Sociales e Igualdad. 2017. Available in: https://portal.guiasalud.es/gpc/lactancia-materna/

(**) Baby-Friendly Hospital Initiative. Ten steps to successful breastfeeding, from UNICEF and the World Health Organization. Available in: https://www.unicef.org/documents/baby-friendly-hospital-initiative

Authors’ response:

We thank the reviewer for reviewing our manuscript and for highlighting references. In the revised manuscript, we have included the suggested additional references [references #40, #41].

2)  Page 2: The text of lines 58 and 59 could be deleted since it repeats what was already expressed in lines 55-57.

Authors’ response:

We agree with the reviewer. We have deleted the repetitive sentence, “As such, DLII may inhibit successful breastfeeding and, in turn, adversely affect both maternal and child health.”

3)  Page 2, line 78: Explain why more than 10 years have passed since the collection of the data (April 2009 – July 2010) until its analysis.

Authors’ response:

This was a secondary analysis using existing data to address a new research question. We have previously published findings from this cohort [References #26-31] In response to this comment, we have inserted the following phrase under materials and methods, study design and population, “The findings from this cohort have been previously published [26-31]. For the current secondary analysis, we analyzed data ...”

4)  Page 2, lines 86-88, 94-96 and figure 1. Did all the women participating in the study have a glucose challenge test (GCT) with a glucose concentration greater than or equal to 7.8 mmol/L after one hour of ingesting 50 g of glucose?. It is indicated that this was the criterion for referred the patient for the 3-hour oral glucose tolerance test (lines 94-96) and that 56 mothers with incomplete 3-hour OGTT data (lines 86-87) were excluded from the study. This information should be clarified.

Authors’ response:

In the manuscript, we have stated that “In clinical practice, the OGTT would generally only be ordered if the GCT was abnormal, whereas in this study the baseline pregnancy OGTT was completed in all participating women.” Study blood samples were collected during OGTT. Therefore, metabolic variables / key exposure variables in the current secondary analysis can only be generated among those with blood samples collected during OGTT. We have further clarified in the revised manuscript by stating “oral glucose tolerance test (OGTT) data (i.e. missing metabolic variable data)”.

5)  Page 2, lines 90-92: If the weight and height measurements were taken during pregnancy, how were you able to calculate the pregravid BMI? Was it really the pregravid BMI or the BMI in the first months of pregnancy?

Authors’ response:

Thank you for this comment. We have revised the following sentence under materials and methods, exposure variables, “Pregravid BMI was calculated [weight (kg)/height (m2)] using the self-reported pregravid weight and the standardized height measures collected at baseline.”

6)  Page 3. Exposure variables (lines 130-148).

Why were the use of anesthesia and the type of anesthesia during delivery not included among the exposure variables?. Previous studies show that it could delay the onset of lactogenesis II (&) The inclusion of this variable would allow comparing the effect of general anesthesia versus epidural anesthesia on the onset of lactogenesis II.

(&) Lind JN, Perrine CG, Li R. Relationship between Use of Labor Pain Medications and Delayed Onset of Lactation. J Hum Lact. 2014

Kutlucan L, Seker Ä°S, Demiraran Y, Ersoy Ö, Karagöz Ä°, Sezen G, Köse SA. Effects of different anesthesia protocols on lactation in the postpartum period. J Turk Ger Gynecol Assoc. 2014 Dec 1;15(4):233-8.

Kocaöz FÅž, Destegül D, Kocaöz S. Comparison of the breastfeeding outcomes and self-efficacy in the early postpartum period of women who had given birth by cesarean under general or spinal anesthesia. J Matern Fetal Neonatal Med. 2019 Jul 10:1-5

It is not indicated how the variable "time to first breast contact" was registered. If it came from the hospital clinical database or from the interviewer-administered questionnaires.

Authors’ response:

We thank the reviewer for providing this perspective. The use and type of anesthesia during delivery were not collected for the original study and, therefore, not available for analysis in the present study. We agree with the reviewer that this is an important point, and therefore we have further discussed this limitation under the discussion section in addition to including recommended references. “Third, use and type of anesthesia were not included among our exposure variables. The original study upon which this secondary analysis is based preceded the literature on the potential effect of general anesthesia versus epidural anesthesia on onset of lacto-genesis II [42-44]. These data were therefore not collected and unable to be considered in the present study.”

Further, the ascertainment of “time to first breast contact” was collected through interviewer-administered questionnaires and is described in detail in the Timing of Lactogenesis II sub-section.

7)  Table 1 should include p-values from bivariate analyses.

Authors’ response:

The requested p values are shown in the table below. Since Table 1 represents the descriptives, we prefer not to include formal analysis and p-values. However, if the reviewer feels strongly, we will revise the manuscript.

Risk factor

DLII - Yes

n (%) or mean ± SD

DLII - No

n (%) or mean ± SD

P

Age (y)

34.7 ± 4.5

35.0 ± 4.2

0.7193

Race

      White

      Non-white

53 (54.1)

45 (45.9)

50 (63.3)

29 (36.7)

0.3115

Pregravid BMI (kg/m2)

24.8 ± 4.6

25.0 ± 5.5

0.8672

Family history of T2DM

      Yes

      No

53 (54.1)

45 (45.9)

42 (53.2)

37 (46.8)

0.9032

Gestational diabetes

      Yes

      No

24 (24.5)

74 (75.5)

16 (20.3)

63 (79.7)

0.5029

Serum concentrations at OGTT

      Fasting glucose (mmol/L)

      HOMA-IR

      ISogtt1

      Adiponectin

4.7 ± 0.6

2.3 ± 1.4

10.9 (8.3, 15.8)

7.1 ± 2.4

4.7 ± 0.9

2.3 ± 1.6

13.0 (8.3, 18.2)

7.4 ± 2.2

0.7589

0.9713

0.3913

0.3694

Length of gestation (wk)

39.3 ± 1.2

39.0 ± 1.2

0.0944

Primiparous

      Yes

      No

60 (61.2)

38 (38.8)

33 (41.8)

46 (58.2)

0.0100

Mode of delivery

      Scheduled cesarean delivery

      Unscheduled cesarean delivery

      Spontaneous

10 (10.2)

31 (31.6)

57 (58.2)

17 (21.8)

12 (15.4)

49 (62.8)

0.0132

Infant birth weight (kg)

3.4 ± 0.5

3.4 ± 0.5

0.9726

Time to first breast contact (hr)

3.8 ± 7.0

2.3 ± 4.5

0.1086

Time to onset of lactogenesis II (hr)

98.3 ± 30.1

57.2 ± 11.7

< 0.0001

8)  In the logistic regression analyzes (Tables 2, 3 and 4), the need for various models is not adequately justified. The last logistic regression model (table 4) simultaneously adjusting for metabolic and obstetric measures, including those of model 1 (only adjusted for age and race) and model 2 (adjusted for age and race, already included in model 1, and primiparity and pregravid BMI ).

Authors’ response:

The Statistical Analyses sub-section has been revised to better justify the logistic regression models used, “The following models were constructed based on the findings of previous studies, including variables of biological and obstetrical relevance. Model 1 was adjusted for age and race. Model 2 was additionally adjusted for primiparity and pregravid BMI. We also performed analysis for a model which simultaneously adjusted for the aforementioned variables (age, race, primiparity, and pregravid BMI) as well as for potential confounders identified in the bivariate analysis (P ≤ 0.10) (mode of delivery and time to first breast contact).”

9)  Page 8,  lines 253-254.  The phrase “This finding is likely because the current cohort population is at higher risk for T2DM” should be explained, and information on this characteristic of the sample should be included in the “Study Design and Population” of the Material and Methods section.

Authors’ response:

This sentence has been revised to better explain the study population characteristics, “This finding may be because the original study was designed to investigate risk factors for T2DM. As such, many individuals at higher risk for T2DM participated in the study although at risk for T2DM was not an inclusion criterion.”

10)  The text of this section is very extensive. It should be summarized in a paragraph.

Authors’ response:

We have revised the section.

Reviewer 2 Report

This is an analysis of data from a cohort study with the aim of determining predictors of delayed lactogenesis. Overall, the study is informative and well written. I have one major concern, which is the limited ability to examine physiological contributions to delayed lactogenesis if there is poor management of breastfeeding in the early postpartum in the study cohort. This reviewer is unclear as to whether formula use during the maternity stay is an available variable. The data set is rich in metabolic variables, but the current approach dilutes the extent to which these variables add to new knowledge. See details below.

  1. Title: Better to use 'secretory activation' or symptoms of 'secretory activation' or 'lactogenesis.' I think 'milk coming in' is too informal for a research paper title.

  1. Abstract: Consider starting abstract with a brief definition of lactogenesis or secretory activation. 'Secretory activation is the onset of copious milk production within a few days of giving birth.'

  1. Line 55: probably more accurate to just include the latter part of this sentence (‘…associated with higher risk for early bf cessation.’) as there isn't evidence that these women are 'often unable to sustain breastfeeding.'

  1. Line 176: Before you report timing of lactogenesis, it is important to report breastfeeding behavior of the cohort. If an appreciable proportion were supplementing with formula in the first 1-2 days, this will override any effect of maternal physiology. While breastfeeding behaviors are important to understand in relation to timing of lactogenesis, to understand the influence of maternal physiology, the cohort needs to have a high intensity of breastfeeding and low intensity of supplementation.

  1. Line 188: Along the same lines as above, in mothers who were exclusively breastfeeding, did you see an effect of metabolic variables after adjusting for parity? Or did you see an interaction with parity in those exclusively breastfeeding?

  1. Line 258. Here, you sort of touch on the concern that I express in comments 4 and 5 above. If there is a high rate of supplementation and many mothers are not breastfeeding frequently in the early postpartum, then breastfeeding management factors are going to override physiology. As an extreme example, it is not possible to study physiologic factors associated with low milk supply if mothers early on decide they only want to breastfeed 1-2 times per day. Poor breast drainage will override the ability to detect physiology. I don’t know if poor breastfeeding management is common in your cohort because you only report on timing of initiation. However, it is important to consider that delayed first breastfeeding and cesarean delivery could be representing overall poor breastfeeding practices in the early postpartum, and these significant factors in your model are not the direct causal factors.

Author Response

Reviewer 2:

Comments:  This is an analysis of data from a cohort study with the aim of determining predictors of delayed lactogenesis. Overall, the study is informative and well written. I have one major concern, which is the limited ability to examine physiological contributions to delayed lactogenesis if there is poor management of breastfeeding in the early postpartum in the study cohort. This reviewer is unclear as to whether formula use during the maternity stay is an available variable. The data set is rich in metabolic variables, but the current approach dilutes the extent to which these variables add to new knowledge. See details below.

Authors’ response:

We thank the reviewer for reviewing our manuscript and for this helpful comment.

1)  Title: Better to use 'secretory activation' or symptoms of 'secretory activation' or 'lactogenesis.' I think 'milk coming in' is too informal for a research paper title. 

Authors’ response:

The title has been revised to include a more formal term, “lactogenesis II’.

2)  Abstract: Consider starting abstract with a brief definition of lactogenesis or secretory activation. 'Secretory activation is the onset of copious milk production within a few days of giving birth.'

Authors’ response:

The abstract has been revised to include a definition of lactogenesis.

3)  Line 55: probably more accurate to just include the latter part of this sentence (‘…associated with higher risk for early bf cessation.’) as there isn't evidence that these women are 'often unable to sustain breastfeeding.'

Authors’ response:

We agree with the reviewer that this sentence was repetitive, and therefore, we have deleted in the revised manuscript under Introduction.

4)  Line 176: Before you report timing of lactogenesis, it is important to report breastfeeding behavior of the cohort. If an appreciable proportion were supplementing with formula in the first 1-2 days, this will override any effect of maternal physiology. While breastfeeding behaviors are important to understand in relation to timing of lactogenesis, to understand the influence of maternal physiology, the cohort needs to have a high intensity of breastfeeding and low intensity of supplementation.

Authors’ response:

We thank the reviewer for this helpful comment. We agree with the reviewer and recommend that the future studies to further examine the influence of supplementation in detail. This is a limitation of the current study and have inserted this recommendation under the revised Discussion section. “Finally, examining the influence of supplementation was not an objective of the present study. Future studies that further examine metabolic and obstetric factors among those who breastfeed exclusively compared to those who supplement with formula may enable a more distinctive understanding of the role of maternal physiology and the role of breastfeeding management in DLII.”

5)  Line 188: Along the same lines as above, in mothers who were exclusively breastfeeding, did you see an effect of metabolic variables after adjusting for parity? Or did you see an interaction with parity in those exclusively breastfeeding?

Authors’ response:

We thank the reviewer for this helpful comment. While this topic is beyond the scope of the objectives of the present study, we agree with the reviewer that future studies should further examine the influence of metabolic and obstetric factors among those who breastfeed exclusively compared to those who supplement. We have inserted this recommendation under the revised Discussion section.

6)  Line 258. Here, you sort of touch on the concern that I express in comments 4 and 5 above. If there is a high rate of supplementation and many mothers are not breastfeeding frequently in the early postpartum, then breastfeeding management factors are going to override physiology. As an extreme example, it is not possible to study physiologic factors associated with low milk supply if mothers early on decide they only want to breastfeed 1-2 times per day. Poor breast drainage will override the ability to detect physiology. I don’t know if poor breastfeeding management is common in your cohort because you only report on timing of initiation. However, it is important to consider that delayed first breastfeeding and cesarean delivery could be representing overall poor breastfeeding practices in the early postpartum, and these significant factors in your model are not the direct causal factors.

Authors’ response:

We thank the reviewer for this insight. We agree with the reviewer that these considerations are essential to the understanding of delayed onset of lactogenesis II. We have addressed the need for these considerations under the revised Discussion section.
